# An Analysis of the Dimensional Constructs of Green Innovation in Manufacturing Enterprises: Scale Development and Empirical Testing

**Siqi Li [1], Xintao Li [2,\*], Qingqing Zhao [3], Jun Zhang [1] and Haoyu Xue [4]**

[1] School of Humanities and Social Sciences, Beijing Institute of Petrochemical Technology, Beijing 102627, China
[2] Zhou Enlai School of Government, Nankai University, Tianjin 300350, China
[3] Development Center of Innovation and Entrepreneurship, Tianjin Sino-German University of Applied Sciences, Tianjin 300350, China
[4] School of Political Science, Law and Public Administration, Shaanxi Normal University, Xi'an 710062, China
\* Correspondence: xintao_li@nankai.edu.cn

**Abstract:** Green innovation has become an important way for manufacturing enterprises to achieve sustainable development. Existing research on green innovation mostly focuses on conceptual content, and relevant empirical studies only involve individual indicators. Systematic measurement tools are still lacking. Therefore, drawing on the theoretical framework provided by the social-ecological system, a clear definition of the connotations of green innovation and an exploration of the dimensional structure and scale development of green innovation in enterprises are the core objectives of this study. Strictly adhering to the process of scale development, four core dimensions of green innovation in enterprises are identified, and a scale with 18 items was established. The results show the following: (1) The connotations of corporate green innovation should be expanded to include not only technology and product innovation but also internal institutional innovation and external environmental innovation. (2) Green innovation in enterprises should be measured in a rigorous manner, and such measurements should follow the logic from goals to behavior. (3) The scale of green innovation should be refined, and questions should be designed to characterize specific green innovation behavior indicators. In summary, this study lays a foundation for quantitative research on green innovation in enterprises and provides a useful reference for manufacturing enterprises to improve their level of green innovation.

**Keywords:** green innovation; scale development; factor analysis; manufacturing enterprises

## 1. Introduction

Green innovation has become an important engine for national and regional economic growth and sustainable development [1]. As the main body of the market economy, the level of green innovation of an enterprise reflects its competitiveness and growth stage to a certain extent [2]. Compared with traditional innovation, green innovation emphasizes low pollution, low emissions, low energy consumption, recyclability, etc. For enterprises, the profit return cycle is often long, and the economic benefits are not obvious. Moreover, the investment threshold related to capital and technology is still high, so many enterprises regard green innovation as an additional means to respond to the government's call and shape their image [3]. As such, green innovation has not yet been integrated into the strategic scope for integrated planning and systematic implementation. However, given the relevance of sustainable development, more and more enterprises, especially traditional manufacturing enterprises, are beginning to realize the important

role of green innovation. In addition to responding to government and shaping one's image, green innovation can help enterprises overcome technological barriers, create differentiated products, optimize management systems, and create a good internal and external environment for the enterprise, so as to comprehensively enhance a sustainable competitive advantage [4].

Social-ecological systems refer to the linked systems of people and nature. Such systems are dynamic, unbalanced, and hierarchical [5]. In recent years, the differential impact of social-ecological systems on different forms of green innovation has become a research hotspot [6]. Academic research on green innovation has been conducted regarding its driving forces, management strategies, and social effects. In terms of drivers of innovation and green innovation, cultural contexts provide an important foundation [7]. Researchers found that different dimensions of culture, such as the acceptance of uncertainty, power distance, and individualism, have a significant impact on the level of innovation [8,9]. In terms of managerial strategies, researchers have studied the impact of corporate environmental performance on financial performance, showing that improved environmental performance by enterprises leads to short-term declines in financial performance [10,11], but in the long run, it increases potential value and reduces risk [12–14]. In terms of social impact, technology has the dual effect of displacing and restoring labor [15]. Disembodied technological change turns out to positively affect employment dynamics in "upstream" sectors, while expansionary investment does so in "downstream" industries [16]. In general, green innovation has a significantly positive impact on long-term job creation [17,18].

As research continues to deepen, the connotations of green innovation have expanded to include a three-dimensional structure of products, technology, and society [19] or three levels of technology, society, and institutions [20]. Such analysis involves a qualitative deconstruction of the concept. In quantitative research, existing studies regard only green innovation as green behavior or take only the number of green patent applications and authorizations as the sole indicator of an enterprise's green innovation. Consequently, these studies fail to comprehensively analyze the behavioral diversity and social performance of green innovation in enterprises. The driving factors of green innovation have not been explored, nor has there been an in-depth exploration of the inner forces and pathways of green innovation. To offer a more comprehensive conceptual deconstruction, this study analyzes the connotations and dimensional structure of green innovation. We also construct a systematic conceptual measurement index system based on the previous literature and empirical research. The results of this study can serve as a reference for interpreting the behavioral process and internal mechanism of green innovation.

In view of the fact that green innovation is based on technology and product production, manufacturing enterprises are taken as the main research object. First, research related to green innovation is systematically reviewed to analyze the definitions and dimensions of green innovation, environmental innovation, product innovation, technological innovation, and institutional innovation. Each dimensional measurement item is developed and revised through semi-structured interviews. Subsequently, empirical tests are conducted through questionnaires. Finally, the findings of the study are summarized to illustrate the limitations and future prospects of the study.

The present research contributes to the development of theory and practice from several aspects. First, we advance the understanding of green innovation in an enterprise by identifying the major dimensions based on an extensive literature review. We provide a comprehensive way to measure green innovation from four major dimensions, including green technological innovation, green product innovation, green institutional innovation, and green environmental innovation. Second, we apply the analytical framework of social-ecological systems to the field of green innovation [5]. Third, we provide practitioners with more operable criteria to evaluate the level of green innovation. In addition, through the application of social-ecological system theory, we found that enterprises should build a green ecological environment to achieve sustainable development. This research will

extend the literature by using social-ecological systems theory to mention the organizational climate and synergy. The scales developed in this study can be used to provide an important reference for green transformation practice. The findings of this study suggest that governments should fully recognize the comprehensive and systematic nature of green innovation, focus on the important role of institutional and cultural construction, develop flexible incentive-based measures to improve the motivation of corporate green innovation, and continually enhance the sustainability of corporate green innovation behavior.

## 2. Literature Review

### 2.1. Connotations of Green Innovation in Enterprises

The concept of green innovation was first proposed at the end of the 20th century [21], and its connotation has been gradually clarified since then. Green innovation with regard to enterprises refers to the innovative activities of enterprises adopting software and hardware related to green products or processes, including pollution prevention, energy conservation, environmental protection, waste recycling, green product design, and enterprise environmental management [22]. The core focus of green innovation is on effect-oriented behavioral processes, i.e., environmental protection and conservation behavior in product development and design, production process, marketing and promotion, and technology application, with the goal of reducing environmental pollution and resource consumption in the entire cycle of the organization's business activities [23]. Green innovation is not limited to environmental protection and energy conservation innovations that reduce pollution but includes technological innovation and product design related to environmental management [24]. In later studies, the connotations of green innovation have been further expanded to include more enterprise management behavior, such as institution-building innovation, cultural management innovation, and stakeholder management innovation, and green innovation has been extended to connotations spatially outside of the enterprise, such as ecological innovation and environmental innovation [25]. Long-term effects, such as sustainable innovation, are also emphasized in terms of time [26]. There are two current definitions of green innovation in academic circles. The first defines green innovation as innovation that positively affects the ecological environment and improves environmental performance, with a focus on enterprise technology and process innovation. The other defines green innovation itself as ecological and environmental innovation, focusing more on systems, culture, and organizational construction [27]. Synthesizing the above views, this study further refines the definition of green innovation proposed by Dai and Cantor [28] and proposes that green innovation is the innovation activity of enterprises to reduce environmental pollution, reduce resource consumption, improve production efficiency, and integrate environmental ecology by improving individual aspects, such as manufacturing technology and production process, or the entire management process and organizational system.

Green innovation, in a narrow sense, refers to green technological innovation and green product innovation. Research on new technology and new product-driven green innovation points out that green technology and product innovation research is not limited to the use of new technologies and tools in the innovation process but also includes innovation performance, increased added value, and social gains based on technology use and product generation [29]. This is in line with the sequence of green innovation research from post-effects to behavior. In the context of digital transformation generally carried out in the manufacturing industry, the connotations of green technological innovation can be expanded to new technologies, new crafts, and new processes [30]. Green product innovation can be summarized as new tangible entities, intangible services, and green carriers to promote resource saving and cost reduction, product performance, and value enhancement [31] while highlighting its important role in information sharing and environmental integration [32]. Ultimately, green innovation results in green, safe, and efficient product

and service outputs. Based on this, this study defines green technological innovation and product innovation as the application of new technologies, tools, and methods to replace traditional processes in the main links of the value chain, such as research and design, procurement and supply, manufacturing, marketing, and logistics. It aims to restructure production and operation methods in order to form new subjects or areas such as green products, green supply chain management, green production and manufacturing, and green operation and sales.

Green innovation, in a broad sense, is regarded as a business strategy element that revolves around the core competence elements of enterprises and adds broader content such as green system innovation, green organization and management innovation, and green environment innovation on the basis of green technology and green product innovation. Related studies pay more attention to external factors driving green innovation, such as market demand, environmental protection policies, industrial environment, and stakeholder relationships [33]. Scholars in this field argue that green innovation is a continuous process that requires not only short-term environmental pollution reduction and resource consumption reduction but also long-term production efficiency improvement, management model optimization [34], competitiveness enhancement [35], and ecosystem construction. Therefore, green innovation, in a broad sense, can be defined as the whole process of reshaping the green innovation behavior of enterprises by reconstructing all aspects of innovation elements and innovation subjects, with new technology as the means and new products as the carrier. It is a process of trial and error and iteration to improve the success rate of innovation [36] so as to enhance the implementation of an enterprise green management strategy on the ground and realize the sustainable development of green innovation and value creation.

### 2.2. Dimensional Components of Green Innovation in Enterprises

Green innovation is an important part of the transformational development and sustainable growth practices of manufacturing companies. Rennings proposes three variations of sustainable development: technological innovation, social innovation, and institutional innovation [20]. Based on this, Hellstrom argues that achieving green innovation needs to consider three aspects: technology, society, and institutions [21]. As a result, it is necessary for enterprises to consider green innovation and sustainable development from technological, social, and institutional aspects. In particular, a product (or service) is the main purpose and core result of an enterprise's technological output. It significantly reflects the degree of green innovation and should be extracted from technological innovation as a dimension for in-depth study. Therefore, this study divides green technological innovation into four dimensions: green technological innovation, green product innovation, green institutional innovation, and green environmental innovation (see details in Table 1).

**Table 1.** Mapping of Green Innovation Dimensions.

| No. | Study | Classification of Green Innovation | | | |
|---|---|---|---|---|---|
| | | Technological Innovation | Product Innovation | Institutional Innovation | Environmental Innovation |
| 1 | Rennings K [20] | ● | | ● | ● |
| 2 | Beise M, Rennings K [37] | ● | | ● | |
| 3 | Frondel M, Horbach J, et al. [38] | | | | ● |
| 4 | Noppers EH, Keizer K, et al. [39] | ● | ● | | |
| 5 | Carrion-Flores CE, Innes R [40] | | | | ● |
| 6 | Abdul-Nasser, El-Kassar, et al. [41] | ● | | | |
| 7 | Cainelli G, Mazzanti M, et al. [42] | | | | ● |
| 8 | Brunnermeier B, Cohen A [43] | | ● | ● | |

### 2.2.1. Green Technological Innovation

With the goal of reducing environmental pollution and saving energy and resources, green innovation in enterprises involves the improvement of technology and processes. On the one hand, the replacement of traditional technologies with new technologies can bring added environmental benefits, such as the adoption of environmentally friendly waste treatment or recycling processes by enterprises. This can effectively reduce environmental pollution emissions and improve resource utilization efficiency [44]. On the other hand, the development and reserve of new technologies can also reverse the influence on the conversion of organizational management policy and enhance green concepts and awareness. Many scholars use the number of green patents granted as a measure of green innovation, which is a widely recognized and adopted proxy indicator of green innovation in academia [2]. Considering that there is a certain lag in the granting of green patents, the more stable, reliable, and timely number of green patent applications is also used as an important indicator in relevant empirical studies [45].

### 2.2.2. Green Product Innovation

The marketability of the product is the ultimate presentation of the technology and the ultimate goal of the activity of the production system of the company. The European Commission defines green product innovation as products that reduce negative impacts and risks to the environment, use fewer resources, and prevent the generation of waste at the product disposal stage [46]. Faced with the widespread problems of environmental pollution and resource consumption, green product innovation is one of the important ways to solve those problems. As a subject of the market economy, enterprises develop green products that are able to contribute to the goal of environmental sustainability [47]. In green innovation practice, enterprises can realize green product innovation in terms of greening product design and development, product green certification, and green product investment, while the market performance of green products (e.g., market share) can characterize the degree of green product innovation in enterprises [48].

### 2.2.3. Green Institutional Innovation

Enterprises are profitable organizations, and in order to market their products and gain more market share and profits, they need to achieve non-technological innovation in terms of systems, models, and management styles. Institutions, as the support system for enterprise innovation, can provide guidance and guarantees for enterprise green behavior [20]. In traditional studies, many scholars have conducted green institutional research in the larger context of countries, regions, and industries, exploring the regulation of green behavior at the policy and regulation levels, and have not yet focused on individual organizations. In contrast, green institutional innovation, which is one of the dimensions of green innovation, is more clearly defined as the level of enterprise institutional construction related to green technologies and green products, involving self-developed green technology standards, the transformation of green technological achievements, energy structure optimization, total environmental quality management, and the construction and implementation of related regulations [49].

### 2.2.4. Green Environmental Innovation

Green innovation is not confined to the interior of the enterprise, as technologies, products, and institutions cannot fully explain all its connotations. Studies have shown that green innovation has spatial externalities, concentrating on environmental spillovers [50]. Therefore, some scholars have argued that green innovation is equivalent to eco-innovation, sustainable innovation, and environmental innovation, among others [51]. This study defines green environmental innovation as a dimension of enterprise green innova-

tion, emphasizing that this externality and spillover is shown through the direct performance of enterprise green innovation as a measure of the degree and contribution of enterprise green innovation.

### 2.3. Social-Ecological Systems Theory

As our understanding of human–environment interactions (HIEs) grows, the exploration of collective human action increasingly requires the inclusion of social and natural ecological environments as important factors in relevant analyses. Ostrom proposed the Social-Ecological Systems (SES) framework to enable researchers to examine the interactions between variables in complex social-ecological systems and the impact of these complex interrelationships on collective action outcomes [52]. The SES framework is a general analytical framework composed of variables at multiple levels. Figure 1 presents the first level of variables in the SES framework. In the first level of the SES framework, four subsystems—resource system (RS), resource unit (RU), governance system (GS), and actor (A)—collectively influence the process of interaction in the action arena (I) and the outcome of collective action (O) [53].

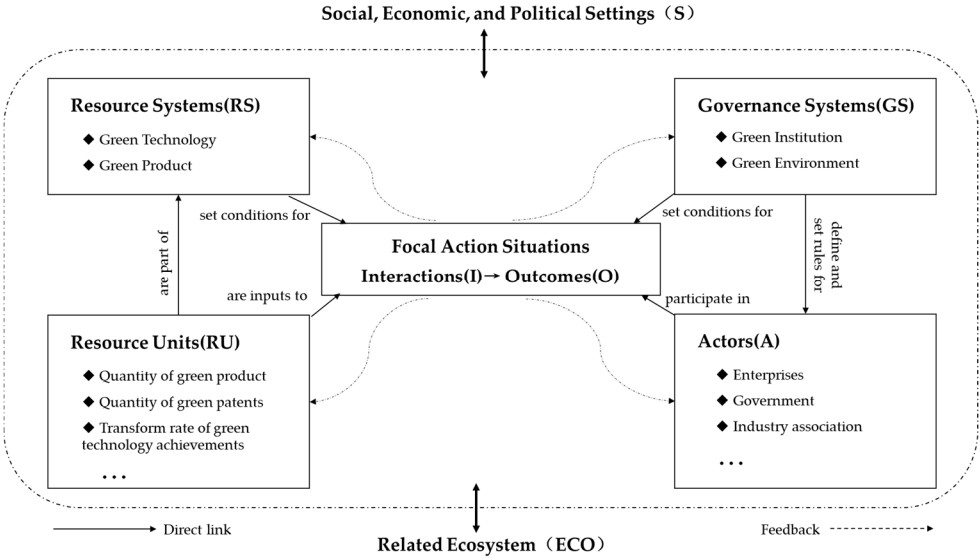

**Figure 1.** Green innovation based on social-ecological systems.

To apply the social-ecological system theory, the dimensions of corporate green innovation can be grouped into this analytical framework (see details in Figure 1). The resource system (RS) constitutes the innovation vehicle of the enterprise, which contains green technology and green products. The governance systems (GS) assume a rule-making function involving green institutions and a green environment. The resource unit (GU) is used for basic measurements, such as the number of green products, the number of green patents, etc. Social ecosystems emphasize the synergistic evolution of multiple actors (GU). Among these actors, this study focuses on companies in stakeholder relationships. The building of "green innovation" is an effective way to hasten the development of the economy and society by considering both natural and social ecosystems and guaranteeing sustained development.

### 2.4. Research Review

To sum up, numerous scholars have conducted rich research on green innovation and social-ecological systems, but there are also deficiencies.

Studies have been conducted to clearly define the conceptual connotations of green innovation, and scholars generally agree that enterprise green innovation is an innovative

action for the improvement of environmental protection and resource utilization. Therefore, in terms of dimension development, more scholars tend to regard green innovation as the improvement actions of enterprises on products and technologies and processes. Technologies and products tend to be taken as its sub-dimensions. However, scholars do not agree on the conceptual extension of enterprise green innovation, and whether to recognize the systemic and external nature of innovation, i.e., innovation as internal management innovation action and external environment construction action. This has become the focus of debate. This also brings about uncertainty in the structure and ambiguity to the measurement of the dimensions of enterprise green innovation. In addition, previous research lacked the combination of key indicators, including the whole process of innovation. In view of the gradual increase of green behavior by traditional enterprises in the process of green transformation, measuring the green innovation level has become one of the important indicators to study the sustainable growth of enterprises. As such, it is necessary and urgent to clarify the dimensions of green innovation and develop a scale.

Scholars focus on the qualitative analysis of social-ecological systems and the relationship between innovation and the context while paying less attention to the quantitative analysis of social-ecological systems theory from the perspective of cultural and collaborative innovation. Based on this, drawing on the social-ecological systems theory and the connotation of green innovation, this paper selects the key indicators affecting the green innovation of enterprises from the perspectives of technology, production, and management and conducts an empirical test of these indicators from the perspective of social-ecological systems.

## 3. Research Hypothesis

We referenced existing research methods on scale development [54,55] and systematically developed a scale of green innovation in enterprises. Firstly, the objectives of green innovation were identified as environmental protection, resource utilization, management integration, and competitiveness enhancement. Secondly, the objectives were used to deduce the process and behavior, refine the connotations, dimensions, and related items of green innovation, and initially frame the four dimensions of the concept as green technological innovation, green product innovation, green institutional innovation, and green environmental innovation. Finally, various methods were used to select the items and test the validity of the scale in order to construct a complete dimensional structure of green innovation and scientifically generate a corresponding measurement scale.

Based on the construction of the green innovation scale, we tested the hypothesis to examine the predictive validity of the scale. According to the social-ecological system theory, enterprises build an organizational innovation climate internally and generate a high level of organizational synergy externally in the social ecology [52]. Green innovation, as one of the important innovations of enterprises, has an important role in organizational climate and organizational synergy.

Organizational climate refers to employees' perception of the psychological atmosphere of the working environment [56]. It expresses employees' shared perceptions about the organizational practices, followed processes, and functioning in the organization [57]. Studies on innovative organizational climate focused on measuring innovative climate [58]. Environmental innovation creates a culture of innovation and shared values in the organization, which forms the basis of the organizational climate [59]. Therefore, organizational climate is also considered to be an important outcome variable for green innovation. Based on the above arguments, we hypothesize that:

**Hypothesis 1 (H1)**. *Green innovation positively relates to organizational climate.*

Organizational synergy refers to the capability of the enterprise to support activities. It is a key sustainable weapon in the highly interactive environment. The social-ecological system theory focuses on the synergy between enterprises and government and enter-

prises. Some previous research found that the high level of innovation influences the degree of integration of activities undertaken by enterprises in the innovation process [60]. Thus, green innovation in enterprises leads to more synergistic behavior. Based on the above arguments, we hypothesize that:

**Hypothesis 2 (H2)**. *Green innovation positively relates to organizational synergy.*

## 4. Research Methodology

This study follows the scale development procedure suggested by Churchill (1979) [61]. We developed a novel scale for measuring enterprise green innovation and conducted an empirical test. It consisted of four steps: (1) Item generation; (2) Content Validity Analysis; (3) Exploratory and confirmatory factor analysis; and (4) Predictive Validity Analysis (see details in Figure 2). First, we developed original items based on secondary data and explored the different dimensions of green innovation through semi-structured interviews. After initial processing on the original scale, we refined the items of green innovation through expert review. Next, we conducted a pre-test by distributing the surveys to a small sample. We distributed the surveys to a large sample of staff and managers in green enterprises. Exploratory and confirmatory factor analyses were conducted to provide insights into the quality of the scale. Finally, a hypothesis test was conducted to examine the predictive validity of the developed scales.

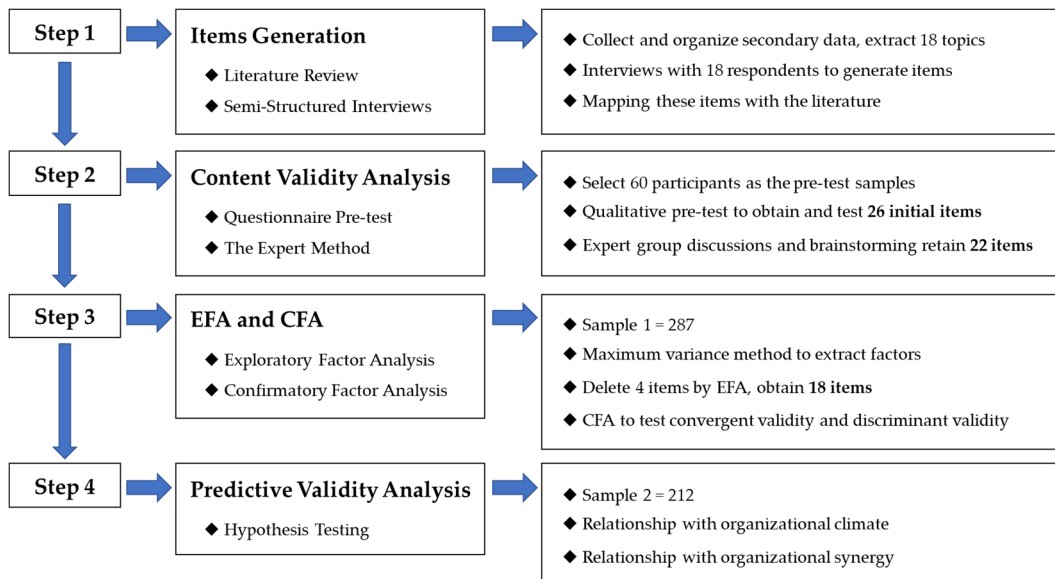

**Figure 2.** Key steps of methodology.

### 4.1. Semi-Structured Interviews

Semi-structured interviews were developed in a two-step process. The first step was to collect and organize secondary data, using "green innovation" and "green transformation" as keywords to search in major databases, collect literature related to dimensional construction and empirical research, and refine the existing topics of green innovation. For example, Camison and Villar-Lopez mentioned "The production process adopted by my company can effectively prevent and abate pollution" as the main criterion to examine green innovation [62]; Sui et al. suggested that "My company's focus on building green supply chains" could be used to characterize the degree of green innovation [49]. Our research team systematically combed through these statements, by listing and summarizing the relevant indicators and criteria, following the causal chain of "behavior–performance" through repeating discussions and deliberations. Ultimately, we extracted 18 topics related to enterprise green innovation from this literature. The second step was to obtain and refine the primary data, explore the different dimensions of green innovation

through semi-structured interviews, and revise and improve the existing measurement items to form the initial scale. The final scale was then formed by testing the reliability and validity of the scale.

*4.2. Questionnaire Pre-Test*

Based on the literature review and preliminary connotation definition, an open-ended questionnaire was designed and distributed to a small sample of Chinese traditional manufacturing companies' senior management teams and managers of technology R&D, production operations, marketing, and other related departments. According to the requirements of the questionnaire survey method, the pre-test sample size was 20–30% of the total sample size. It was estimated that the total number of samples would be about 200–300, so 60 participants were selected as the pre-test samples (N = 60). We randomly selected 3–5 employees or supervisors from each of the 18 manufacturing enterprises in China as the participants.

The participants were asked to describe and give examples of each of the four dimensions (at least three specific behaviors or examples for each dimension) based on the given definition of "green innovation." Finally, 37 questionnaires with 24 descriptions were collected.

The initial question items were extracted from the original data. The principle of extraction was as follows: clear descriptions with concise sentences and single meanings were directly retained as valid questions; for the descriptions with long sentences and multiple meanings, one researcher and the two collaborators discussed and streamlined, split, or eliminated them; descriptions that were ambiguous and not closely related to the connotation structure of green innovation were directly deleted; for the case descriptions, after repeated reading and understanding, their important meanings were refined into the initial items; for the items with repeated meanings, only one item was retained. According to the above principles, eight items were deleted and 16 items were retained.

The 16 initial items obtained by induction were combined with the 18 items obtained by literature research. The principles of merging were as follows: the items obtained by the literature method were retained for topics with the same meaning; the items with intersecting meanings were discussed and rewritten with other researchers; the new items obtained by the inductive method were retained directly; and the number of items in each dimension was kept equal or close to each other as much as possible in order to maintain a balanced structure of the constructs. According to the above principles, the combined result was 26 items: nine items of green technological innovation, five items of green product innovation, six items of green institutional innovation, and six items of green environment innovation (see details in Table 2).

**Table 2.** Initial items of green innovation in enterprises.

| Dimensions | Item No. | Item Contents |
|---|---|---|
| | 1 | The production process adopted by my company can effectively prevent and abate pollution |
| | 2 | The technology adopted by my company can effectively reduce energy consumption |
| | 3 | My company adopts green technology |
| | 4 | My company uses environmentally friendly waste treatment or recycling process |
| | 5 | My company applies for green patents that have been or will be granted |
| Resource system (RS) | 6 | My company's green technology capabilities have been enhanced |
| | 7 | My company attaches importance to investments in green technology |
| | 8 | The technology process adopted by my company has been improved |
| | 9 | All of my company's technology can pass the Green National Technical Certification |
| | 10 | My company considers reducing pollution and saving energy in its product design |
| | 11 | All of my company's products can pass the green product certification |
| | 12 | My company attaches importance to investments in green products |

| | | |
|---|---|---|
| | 13 | My company has increased its market share of green products |
| | 14 | The proportion of green products in my company has been improved |
| | 1 | My company has promulgated green technology standards |
| | 2 | My company has improved the transformation rate of green technology achievements |
| | 3 | My company carries out overall planning to optimize the energy structure |
| | 4 | My company has implemented total environmental quality management |
| | 5 | My company focuses on green-related regulations, construction, and cultural promotion |
| Governance systems (GS) | 6 | My company's green-related regulations have been well implemented |
| | 7 | My company is committed to launching green products |
| | 8 | My company is committed to improving the quality of its green services |
| | 9 | My company focuses on building green supply chains |
| | 10 | My company focuses on improving green marketing performance |
| | 11 | My company has created a good industrial ecology |
| | 12 | My company feels responsible for the green environment |

### 4.3. The Expert Method

In order to ensure the content validity of the original scale, an expert group was established composed of doctoral students, lecturers, associate professors, and professors from well-known universities and research institutes (e.g., Renmin University of China and Peking University) with a certain research base in the fields of green innovation, the green transformation of manufacturing industries, and the sustainable development of enterprises. We provided all the experts with a short reference list, our conceptualization of enterprise green innovation, and some considerations about item development. The experts were asked to check the suitability of items we generated previously. The experts reviewed the items one by one in a back-to-back manner. Through group discussions and brainstorming, 26 green innovation items obtained from previous literature research and open-ended questionnaires were revised; two items (GT7 and GT8) for green technological innovation, one item (GI6) for green institutional innovation, and one item (GE6) for green environmental innovation were deleted, retaining 22 items.

### 4.4. Exploratory and Confirmatory Factor Analyses

After forming the 22-item scale, a questionnaire survey was conducted to test the scale empirically. In this study, questionnaires were distributed from January to September 2022 using various channels, such as WeChat, the wjx.cn platform, and email, mainly targeting executives and managers of R&D, production, and other related departments of Chinese traditional manufacturing enterprises. The average age was 41.49 years old in the sample, and the participants comprised those with the right to decide or implement green innovation decisions and actions in their enterprises and master the details of green innovation. The industry distribution of these enterprises was 37.6% in resource processing, 33.4% in machinery and electronic manufacturing, and 29.0% in light textiles. Samples were all traditional manufacturing enterprises and had strong sample representativeness. A total of 308 questionnaires were distributed, 287 were collected, and 23 of the disqualified questionnaires were excluded, with a valid recovery rate of 85.71%.

Exploratory and confirmatory factor analyses were conducted for this sample to test the reliability of the scale. We ran exploratory factor analysis (EFA) to test the factorial structure and validate the scale. It attempted to identify factors that explained the pattern of correlations within a set of observed variables. EFA was conducted based on varimax rotation and principal component analysis using SPSS 22. Two main indicators showed the results. The first indicator, Kaiser-Meyer-Olkin (KMO), indicated that the sample was appropriate for factor analysis. The second indicator, Bartlett's value, showed the data's suitability for factor analysis. Next, we performed a confirmatory factor analysis (CFA) to check the factorial structure and verify the measurement model. The CFA was conducted

based on convergent and discriminant validity analysis using Amos 24, used to test how well the measured variables represent the number of constructs.

### 4.5. Predictive Validity Analysis

We collected data from a new sample—five companies in China covering the petrochemical industry, the electric power industry, and the textile industry. After receiving an email containing information about our study, participants were requested to complete our online questionnaire via a survey link. The average age of participants was 42.21 in the new sample. The industry distribution of these enterprises was 38.7% in Petrochemistry, 30.3% in electric power, and 31.0% in textiles. Samples were all traditional manufacturing enterprises and had strong sample representativeness. A total of 256 questionnaires were distributed, 212 were collected, and 44 of the disqualified questionnaires were excluded, with a valid recovery rate of 82.81%.

All constructs in the model were measured with multiple-item scales. Each of these variables was measured by a five-point Likert-type scale, ranging from 1 (strongly disagree) to 5 (strongly agree). The organizational climate was measured using the 8-item scale developed by Scott and Bruce (1994) [58]. Results of Cronbach's alpha ($\alpha = 0.905$) showed the high reliability of the scale. Organization synergy measured by four items was adapted with some modification in words from Augusto and Coelho (2009) [63]. Cronbach's alpha results ($\alpha = 0.903$) showed the high reliability of the scale.

## 5. Results

### 5.1. Exploratory Factor Analysis

We conducted EFA to examine the factorial structure of the 22 items. The Kaiser–Meyer–Olkin (KMO) value for the data collected in this study was 0.850, indicating that the data were appropriately sampled. Additionally, Bartlett's test ($\chi 2 = 2567.809$, $p < 0.001$) was conducted, indicating that the variables were suitable for factor analysis [64].

Using the maximum variance method to extract factors with eigenvalues greater than 1 from the 22 question items of green innovation, a total of four factors were extracted with a cumulative variance explained of 58.431%, i.e., more than 50%. Most of the question items had factor loadings between 0.600 and 0.892, among which four question items (GT6, GT7, GP5, and GE5) had factor loadings lower than 0.5. After deleting these four question items, the maximum variance method was reselected for the analysis, as shown in Table 3. The results showed that the corresponding factor loadings of each entry were greater than 0.6, and the cumulative variance explained was 67.790%, which was significantly higher than the explanation rate before removal and greater than 60%, indicating a more desirable factor structure. The Cronbach's $\alpha$ coefficients of each factor obtained after exploratory factor analysis were all greater than 0.7, indicating that the scale had good stability [65], as shown in Tables 4 and 5.

**Table 3.** Results of exploratory factor analysis (N = 264).

| Items | Factor Loadings | | | |
|:-----:|:----:|:----:|:----:|:----:|
| | **1** | **2** | **3** | **4** |
| GT4 | 0.870 | | | |
| GT2 | 0.837 | | | |
| GT1 | 0.831 | | | |
| GT3 | 0.796 | | | |
| GT5 | 0.705 | | | |
| GP2 | | 0.892 | | |
| GP1 | | 0.873 | | |
| GP3 | | 0.861 | | |
| GP4 | | 0.802 | | |

| | 1 | 2 | 3 | 4 |
|---|---|---|---|---|
| GI2 | | | 0.791 | |
| GI1 | | | 0.781 | |
| GI5 | | | 0.779 | |
| GI3 | | | 0.734 | |
| GI4 | | | 0.710 | |
| GE2 | | | | 0.817 |
| GE1 | | | | 0.778 |
| GE3 | | | | 0.706 |
| GE4 | | | | 0.600 |
| Eigenvalues (non-rotational values) | 6.038 | 2.562 | 2.120 | 1.483 |
| Explanation rate % (Total 67.790%) | 19.266 | 17.842 | 17.456 | 13.226 |

Notes: GT = green technological innovation; GP = green product innovation; GI = green institutional innovation; GE = green environmental innovation.

**Table 4.** Scale and source of green innovation measurement.

| Variables | Item Contents | Source |
|---|---|---|
| Green Technological Innovation (GT) | The production process adopted by my company can effectively prevent and abate pollution (GT1) | Camison et al., 2014 [62] |
| | The technology adopted by my company can effectively reduce energy consumption (GT2) | Schiederig et al., 2012 [27] |
| | My company adopts green technology (GT3) | First-hand data |
| | My company uses environmentally friendly waste treatment or recycling process (GT4) | Schiederig et al., 2012 [27] |
| | My company applies for green patents which have been or will be granted (GT5) | Qi et al., 2018 [46] |
| Green Product Innovation (GP) | My company considers reducing pollution and saving energy in its product design (GP1) | First-hand data |
| | All of my company's products can pass the green product certification (GP2) | Camison et al., 2014 [62] |
| | My company attaches importance to investments in green products (GP3) | First-hand data |
| | My company has increased its market share of green products (GP4) | Sui et al., 2015 [49] |
| Green Institutional Innovation (GI) | My company has promulgated green technology standards (GI1) | First-hand data |
| | My company has improved the transformation rate of green technology achievements (GI2) | Xiao et al., 2021 [50] |
| | My company carries out overall planning to optimize the energy structure (GI3) | Xiao et al., 2021 [50] |
| | My company has implemented total environmental quality management (GI4) | Schiederig et al., 2012 [27] |
| | My company focuses on constructing green-related regulations and cultural promotion (GI5) | First-hand data |
| Green Environmental Innovation (GE) | My company is committed to launching green products (GE1) | First-hand data |
| | My company is committed to improving the quality of its green services (GE2) | First-hand data |
| | My company focuses on building green supply chains (GE3) | Sui et al., 2015 [49] |
| | My company focuses on improving green marketing performance (GE4) | Sui et al., 2015 [49] |



**Table 5.** Results of the reliability test of the scale (N = 264).

| Variables | Standardized Cronbach's Alpha |
|---|---|
| GT | 0.884 |
| GP | 0.914 |
| GI | 0.839 |
| GE | 0.776 |

### 5.2. Confirmatory Factor Analysis

In this study, confirmatory factor analysis was conducted using Amos 24.0 for the above questions, and the convergent validity was tested using the average variance extracted (AVE), combined reliability (CR), and discriminant validity. Initially, we established a four-factor model based on our conceptualization of enterprise green innovation. Specifically, four latent variables were created: green technological innovation (5 items), green product innovation (4 items), green institutional innovation (5 items), and green environmental innovation (4 items).

Before the convergent validity test, the overall scale fit was examined. According to Bentler's suggested criteria [66], $X^2/df \leq 3$, IFI $\geq 0.9$, TLI $\geq 0.9$, CFI $\geq 0.9$, and RMSEA $\leq 0.08$ indicated that the model fit was acceptable, as shown in Table 6.

**Table 6.** Goodness of fit of the scale.

| $\chi^2/df$ | RMSEA | IFI | TLI | CFI |
|---|---|---|---|---|
| 2.292 | 0.070 | 0.934 | 0.920 | 0.933 |

In accordance with the suggestion of Fornell and Larcker [67], the convergent validity of the scale was determined by the AVE value in this study. The AVE value of the green innovation scale was greater than 0.5, and the reliability of the combination of the four factors of green innovation was greater than 0.8 [68], indicating that the scale has high credibility and stability, as shown in Table 7.

**Table 7.** Results of the convergent validity test.

| | Route | | Estimate | AVE | CR |
|---|---|---|---|---|---|
| GT4 | <--- | GT | 0.812 | | |
| GT3 | <--- | GT | 0.756 | | |
| GT2 | <--- | GT | 0.861 | 0.656 | 0.905 |
| GT1 | <--- | GT | 0.845 | | |
| GT5 | <--- | GT | 0.621 | | |
| GP4 | <--- | GP | 0.772 | | |
| GP3 | <--- | GP | 0.821 | 0.736 | 0.917 |
| GP2 | <--- | GP | 0.922 | | |
| GP1 | <--- | GP | 0.888 | | |
| GI4 | <--- | GI | 0.644 | | |
| GI3 | <--- | GI | 0.665 | | |
| GI2 | <--- | GI | 0.791 | 0.577 | 0.872 |
| GI1 | <--- | GI | 0.766 | | |
| GI5 | <--- | GI | 0.712 | | |
| GE4 | <--- | GE | 0.661 | | |
| GE3 | <--- | GE | 0.721 | 0.533 | 0.818 |
| GE2 | <--- | GE | 0.668 | | |
| GE1 | <--- | GE | 0.685 | | |

Notes: CR = composite reliability; AVE = average variance extracted; GT = green technological innovation; GP = green product innovation; GI = green institutional innovation; GE = green environmental innovation.

The correlation among the dimensions was significant ($p < 0.01$), and the square root of the AVE was greater than the correlation among the sub-dimensions, indicating that the variables had moderate to low correlation. They had common attributes and, at the same time, had their own independence, indicating that the dimensions that make up green innovation had a certain correlation and good differentiation validity, thus forming an organic whole, as shown in Table 8.

**Table 8.** Results of the discriminant validity test.

| | M | SD | GT | GP | GI | GE |
|---|---|---|---|---|---|---|
| GT | 4.328 | 1.124 | **0.810** | | | |
| GP | 5.278 | 1.069 | 0.263 ** | **0.858** | | |
| GI | 5.496 | 0.760 | 0.264 ** | 0.336 ** | **0.760** | |
| GE | 4.593 | 0.973 | 0.363 ** | 0.431 ** | 0.348 ** | **0.730** |

Note: AVEs are reported in bold along the diagonal, ** $p < 0.01$; GT = green technological innovation; GP = green product innovation; GI = green institutional innovation; GE = green environmental innovation.

*5.3. Predictive Validity Analysis*

To test Hypotheses 1 and 2, we first examined correlations between green innovation, organizational climate, and organizational synergy. Green innovation was positively related to enterprise culture ($r = 0.507$, $p < 0.01$). Moreover, green innovation was positively related to enterprise performance ($r = 0.503$, $p < 0.01$) (see details in Table 9).

Next, we conducted regression models, setting green innovation and its four dimensions as the independent variable (see details in Table 10). In support of Hypothesis 1, the regression model showed a positive effect of green innovation on organizational climate ($\beta = 0.641$, $p < 0.01$). All four dimensions of green innovation had positive impacts on organizational climate ($\beta_1 = 0.141$, $p < 0.05$; $\beta_2 = 0.117$, $p < 0.05$; $\beta_3 = 0.303$, $p < 0.01$; $\beta_4 = 0.038$, $p < 0.05$). In line with Hypothesis 2, the regression model showed a positive effect of green innovation on enterprise performance ($\beta = 0.717$, $p < 0.01$). All four dimensions of green innovation had positive impacts on organizational synergy ($\beta_5 = 0.186$, $p < 0.05$; $\beta_6 = 0.078$, $p < 0.05$; $\beta_7 = 0.410$, $p < 0.01$; $\beta_8 = 0.040$, $p < 0.05$). In conclusion, the results strongly support the predictive validity of the previously developed scale.

**Table 9.** Means, standard deviation, and Pearson correlation matrix (N = 212).

| Construct | M | SD | 1 | 2 | 3 |
|---|---|---|---|---|---|
| Green innovation | 3.92 | 0.67 | 1.000 | | |
| Organizational climate | 3.77 | 0.83 | 0.507 ** | 1.000 | |
| Organizational synergy | 3.27 | 0.95 | 0.503 ** | 0.508 ** | 1.000 |

Note: ** $p < 0.01$.

**Table 10.** Regression models of green innovation in enterprise.

| Variables | DV1 = Organizational climate | | DV2 = Organizational synergy | |
|---|---|---|---|---|
| | β | SE | β | SE |
| GT | 0.141 * | 0.078 | 0.186 * | 0.088 |
| GP | 0.117 * | 0.078 | 0.078 * | 0.087 |
| GI | 0.303 ** | 0.087 | 0.410 ** | 0.098 |
| GE | 0.038 * | 0.069 | 0.040 * | 0.078 |
| Green innovation | 0.641 ** | 0.064 | 0.717 ** | 0.073 |
| $R^2$ | 0.275 | | 0.286 | |
| F e | 17.464 ** | | 18.470 ** | |

Note: * $p < 0.05$, ** $p < 0.01$; GT = green technological innovation; GP = green product innovation; GI = green institutional innovation; GE = green environmental innovation.

## 6. Discussion

### 6.1. Summary of the Findings

This study developed and validated an enterprise green innovation scale by combining green innovation-related connotations and dimensional studies. Following the scale development steps, a four-dimensional enterprise green innovation measurement scale containing 18 questions was developed by integrating literature research, semi-structured interviews, and open-ended questionnaires. The cumulative variance explained rate of the scale was nearly 70%, and the internal consistency reliability of all four sub-dimensions was higher than 0.7, indicating that there was good heterogeneity among the items of the four dimensions of the enterprise green innovation scale. In addition, through validation factor analysis, the CR and AVE indicators were used to test the convergent validity and discriminant validity of the enterprise green innovation scale. The results indicated that the generated scale could effectively measure the level of green innovation in enterprises.

The connotations of green innovation have a systemic structural nature. The empirical results show that in the context of green innovation in Chinese manufacturing enterprises, the connotations of green innovation should be defined from both inside and outside the enterprise. They can be further divided into four dimensions: green technological innovation, green product innovation, green institutional innovation, and green environmental innovation. Technological innovation and product innovation can bring short-term economic benefits and long-term public welfare benefits to enterprises, while institutional and environmental innovation create the internal and external environment for sustainable growth, including the internal management environment and external social-ecological environment. Therefore, the connotations of green innovation should be expanded, measured, and examined comprehensively in order to effectively present the structure and systemic nature of its connotations.

Research on green innovation follows an inverse logical order. In the research process, from the four abstract structural dimensions to the concrete 18 topics, we showed that green innovation is consistent with the causal chain of "behavior–performance" by enterprises. After identifying the key performance objectives of green innovation as environmental protection, resource utilization, management integration, and competitiveness enhancement, the process and behavior were inferred from the objectives to refine the connotations, dimensions, and related topics of green innovation. This highlighted that green innovation is a series of behavioral and result-oriented process unification, and the whole process is goal-oriented and result-oriented behavior shaping. As such, the reverse logic of "goal-result-behavior" should be followed in practice.

The green innovation practices of enterprises highlight the diversity of behavior. We found that the 18 items in the green innovation scale were all specific behaviors, such as production processes and procedures adopted by enterprises that can effectively prevent and reduce pollution, products that can pass green product certification with promulgated green technology standards, commitment to launching green products, etc. The

number and degree of green innovation-related behavior adopted by enterprises confirm the level of green innovation. Therefore, based on the extensive research on green innovation, we proposed that the enterprise green innovation scale should be refined, with the questions implemented to behaviors that characterize specific green innovation behavior indicators. The scale provides operational guidelines for enterprise practices to enhance the level of green innovation.

### 6.2. Policy Implications

This study provides directional guidance for manufacturing enterprises on how to fully exploit green elements and improve green innovation capabilities in the pursuit of sustainable development. First, green innovation is an important way for enterprises to develop sustainably. Enterprises should emphasize the importance of green innovation and action, break from the traditional prejudice that green behavior is pure cost behavior, regard green innovation as beneficial behavior, social behavior, and growth behavior, include it into the strategic management category for layout and promotion, and unify the goals, processes, and results in the construction process to build a green and sustainable development path for enterprises. Second, in the process of green innovation by enterprises, it is necessary to focus not only on tangible initiatives such as technology and products but also on intangible construction such as institutions and the environment. We found that technology and product innovation are the foundation of green innovation, while institutional and environmental innovation are equally important components. Enterprises should carry out integrated design and systematic implementation to combine the two and stimulate comprehensive and complete green innovation behavior and effectiveness. Finally, green innovation behavior can be implemented into a variety of actions, and companies should make them concrete, actionable, and measurable. When developing the scale, our research team identified dozens of green innovation behaviors, especially the 18 question items that constitute the scale. These items point out the actions for green innovation in enterprises and can be used to measure their performance. They also provide important references for enterprises to identify, judge, and assess the degree to which green innovation is promoted and how it unfolds.

This study also provides a basis for government policies on green innovation and environmental sustainability. First, governments should fully recognize the comprehensive and systematic nature of green innovation in enterprises and focus on the important role of institutional and cultural construction. Clear product and technology standards should be developed to create a green innovation atmosphere and enhance the promotion of sustainable development concepts. Second, governments should improve the motivation of enterprises toward green innovation by developing flexible incentive-based measures. In addition to tax incentives and environmental subsidies, special support funds for environmental protection can be provided alongside environmental information and technical support. Finally, governments should combine short-term and long-term efforts and flexibly adjust the ratio and strength of different environmental regulations by different industries and enterprises to continually enhance the sustainability of enterprises' green innovation behavior.

## 7. Conclusions

Green innovation is an important approach to the sustainable development of enterprises. This study showed that green innovation in enterprises is a composite concept that contains four core dimensions: green technological innovation, green product innovation, green institutional innovation, and green environmental innovation. Among them, technological and product innovations are the foundation, while institutional and environmental innovations extend the concept from internal and external perspectives, respectively. In addition, the scale was validated and shown to have good reliability as a quantitative measurement tool for green innovation.

Based on previous research on green innovation, this study followed a standardized procedure to develop a green innovation scale for enterprises that promotes the transformation of green innovation from conceptual exploration to empirical analysis. This scale can be used to understand the mechanism of green innovation in manufacturing enterprises. First, we expanded the connotations of green innovation. In previous studies, the connotations of green innovation involved different focuses and lacked a systematic structure [21]. By using a standardized scale development procedure to develop a green innovation scale with good reliability and validity, we found that green innovation contains four dimensions, namely, green technological innovation, green product innovation, green institutional innovation, and green environmental innovation. Compared with existing scales, green technology and product innovation were expanded to include institutional and environmental innovations. Our scale is more specific and comprehensive in its description. It overcomes the shortcomings of the existing scales to a certain extent while providing a foundation for further empirical research. Second, the behavioral indicators of green innovation were refined. By reviewing literature and conducting in-depth interviews, green behavioral indicators were limited to the product and technology levels, and more attention was paid to the study of drivers and environmental after-effects [69]. Our research team identified specific behaviors of green innovation in manufacturing enterprises, focusing more on process exploration and refining the conceptual dimensions on this basis. These behaviors provided direct evidence for the construction of green innovation indicators, with first-hand information for future case studies. Further, they could serve as a foundation for subsequent studies of green behavior in enterprises. Finally, the research context of green innovation was enriched. Traditional manufacturing enterprises in the Chinese context were selected as research samples to develop a scale that expands the applicable contexts of green innovation and reveals the structural uniqueness and dimensional focus of green innovation in different contexts. At the same time, our study of green innovation behavior in different dimensions can provide guidance for subsequent empirical studies with large samples and an expanded multi-context theory. Moreover, the study can serve as a reference for enterprises to improve their level of green innovation and promote environmental sustainability.

There are some limitations to this study. First, the study sample was restricted to Chinese manufacturing enterprises. Future studies should further validate the scale by considering enterprises in other countries and other industries to enhance the generalizability of the enterprise green innovation scale. Second, since there were differences in the degree of green transformation and innovation among different enterprises, future research can further classify green innovation in different enterprises at different stages of development to improve the relevance of the scale. Finally, although we developed and empirically tested the scale, in the future, case studies can be conducted using rooting theory to make the research data richer and the research conclusions more rigorous. Doing so will provide a more useful reference for increasing green transformation and promoting green innovation while supplementing green innovation theory.

**Author Contributions:** All authors contributed equally to this work. S.L. was responsible for reviewing and editing, validation, supervision, and investigation. X.L. was responsible for reviewing and editing, conceptualization, visualization, software, and methodology. Q.Z. was responsible for validation and investigation. J.Z. was responsible for supervision. H.X. was responsible for validation. All authors have read and agreed to the published version of the manuscript.

**Funding:** This research was funded by the National Social Science Foundation of China (21CZZ007), the Tianjin Social Science Foundation of China (TJGLQN20-001), the China Postdoctoral Science Foundation (2020M670636), the Liberal Arts Development Foundation of Nankai University (ZB22BZ0332), and the R&D Program of Beijing Municipal Education Commission (SM202110017003) awarded to the fourth author (Jun Zhang). URL of the funding website: http://jw.beijing.gov.cn/kyc/ (accessed on 14 December 2022).

**Institutional Review Board Statement:** Not applicable.

**Informed Consent Statement:** Not applicable.

**Data Availability Statement:** Not applicable.

**Acknowledgments:** This study was supported by the National Social Science Foundation of China under grant no. 21CZZ007, the Tianjin Social Science Foundation of China, under grant no. TJGLQN20-001, the China Postdoctoral Science Foundation under grant no. 2020M670636, the Liberal Arts Development Foundation of Nankai University under grant no. ZB22BZ0332, and the R&D Program of Beijing Municipal Education Commission under grant no. SM202110017003, awarded to the fourth author (Jun Zhang). URL of the funding website: http://jw.beijing.gov.cn/kyc/ (accessed on 14 December 2022).

**Conflicts of Interest:** The authors declare that the research was conducted in the absence of any commercial or financial relationships that could be construed as a potential conflict of interest.

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
