# Peer review of "An Analysis of the Dimensional Constructs of Green Innovation in Manufacturing Enterprises: Scale Development and Empirical Testing"

_sustainability, doi:10.3390/su142416919_

Round 1

Reviewer 1 Report

This is a novel paper, providing an operational conceptualization of green innovation through a multi-step methodology including a preliminary extraction of 18 key topics from the relevant literature, a further identification of 16 items from a questionnaire survey, and finally a combination and validation of the previous steps by an expert group, ending in identifying 22 items which demonstrated to be further validated by a factor analysis. While I think the paper is well crafted and proposing an interesting taxonomy, I nevertheless think that the following improvements are necessary.

1.      Sections 1 and 2 fail to fully locate the paper in the relevant literature, since important aspects relevant to green innovation strategies are neglected. Even if such aspects are not directly taken into account in the following analysis, they should be discussed in the introductory sections. Three are the main issues. In terms of drivers of innovation and green innovation, the cultural context is extremely important and should not be neglected in the overall introduction. In terms of managerial strategies, green innovation and green performance may have an important impact on the financial performance of a given company. Finally, in terms of social impact, green innovation may be more labor-friendly and less labor-saving, so contributing to job creation (in contrast with most of other process innovations, see references below). Here some suggestions of references useful to incorporate these three perspectives within the introductory framework of the paper; the references are grouped together according to the three suggested perspectives.

Cultural context:

Hofstede, G. H. (1983). The cultural relativity of organizational practices and theories. Journal of International Business Studies, 14, 75–89.

Hofstede, G., Hofstede, G. J., & Minkov, M. (2010). Cultures and organizations: Software of the mind. New York: McGraw-Hill.

Shane, S. (1993). Cultural influences on national rates of innovation. Journal of Business Venturing, 8, 59–73.

Financial performance:

Alsaifi, K., Elnahass, M., Al-Awadhi, A.M. & Salama, A.  (2022). Carbon disclosure and firm risk: evidence from the UK corporate responses to climate change. Eurasian Business Review, 12, 505–526.

Delmas, M., Nairn-Birch, N., & Lim, J. (2015). Dynamics of environmental and financial performance: The case of greenhouse gas emissions. Organization and Environment, 28(4), 374–393.

Fisher-Vanden, K., & Thorburn, K. (2011). Voluntary corporate environmental initiatives and shareholder wealth. Journal of Environmental Economics and Management, 62(3), 430–445.

Trumpp, C., & Günther, T. (2017). Too little or too much? Exploring ushaped relationships between corporate environmental performance and corporate financial performance. Business Strategy and the Environment, 26(1), 49–68.

Xie, J., Nozawa, W., Yagi, M., Fujii, H., & Managi, S. (2019). Do environmental, social, and governance activities improve corporate financial performance? Business Strategy and the Environment, 28(2), 286–300.

Employment impact:

Acemoglu, D., & Restrepo, P. (2019). Automation and new tasks: How technology displaces and reinstates labor. Journal of Economic Perspectives, 33(2), 3–30.

Dosi, G., Piva, M., Virgillito, M. E., & Vivarelli, M. (2021). Embodied and disembodied technological change: The sectoral patterns of job-creation and job-destruction. Research Policy, 50(4), 104199.

Gagliardi, L., Marin, G. & Miriello, C. (2016). The greener the better? Job creation effects of environmentally-friendly technological change, Industrial and Corporate Change, 25, 779-807.

Staccioli, J., Virgillito, M.E. (2021). Back to the past: the historical roots of labor-saving automation. Eurasian Business Review, 11, 27–57.

2.      The Authors should provide more details on the adopted methodologies, with particular references to the topic analysis of the literature (Section 3.1), the actual questions proposed in the questionnaire (Section 3.2) and the way how the expert focus group has actually worked (Section 3.3)

3.      Although titled “Policy implications”, Section 5.2 is actually discussing managerial implications; indeed, policy implications should be discussed and Section 5.2 should be properly extended, also making stock of the additional dimensions added to the discussion in the introductory sections (see point 1).

Author Response

Dear reviewer and editor:

On behalf of my co-authors, we thank you for your professional comments concerning our manuscript entitled “An analysis of the dimensional constructs of green innovation in manufacturing enterprises: Scale development and empirical testing (ID: sustainability-2035736)”. These comments are all valuable and helpful for revising and improving our paper, as well as an important guiding significance to our research. We have studied each comment carefully and made corrections for your kind consideration again.

Please see the attachment. Many thanks!

In conclusion, we sincerely thank you for your professional comments, and we will keep on making efforts to do it better. And we hope to learn more knowledge from you! If there are any further questions, please email us and we will answer them fully and accurately. Thanks again for all the support!

Looking forward to hearing from you.

Thank you and best regards.

Yours sincerely

Reviewer 2 Report

I commend the authors for coming up with a title that is of increasing importance in contemporary times.

I have the following observations:

1. The title of the study is "An analysis on Dimensional constructs of green innovation in 2 manufacturing enterprises: Scale development and empirical 3 testing"

Should it really be "An analysis on Dimensional constructs . . . " or "An analysis of dimensional constructs . . ."

2. The authors did not anchor the study on any theory. ironically , they claimed that "The findings of this study can facilitate an understanding of the connotations and dimensions of green innovation in enterprises and provide measurement tools and theoretical guidance for related empirical 80 analysis. . ."

3. The authors failed to clearly articulate the contributions of the study

4. I suggest that the authors proofread the work properly

5. What informed a choice of 60 respondents as the sample of the study? How were they selected? Why do the authors think that this sample is adequate? What method did they employ?

6. What do the authors mean by "Content Reliability" in line 9 of the abstract?

7. How did the authors use "Exploratory and validation factor analyses  to test the reliability of the scale" . . . They should explain

8. Since the major analysis conducted was on the instrumentation/validity and reliability, the authors should give detailed explanations here

Author Response

(The authors gave the same response as above.)

Reviewer 3 Report

Dear Authors, 

Thank you for your interesting paper, but I have some comments and suggestions:

Please remove from the abstract the detailed description of the survey; for this, you have another part - methodology. The abstract mentioned that you did such research and then presented the results shortly. 

In my opinion, it is needed 2.21, 2.2.2 and 2.2.3

Please comment on the tables.

Author Response

(The authors gave the same response as above.)

Round 2

Reviewer 1 Report

The authors have implemented the suggested changes

Reviewer 2 Report

A good effort. please see the attached file for detailed comments

Round 3

Reviewer 2 Report

A good response but there are still some issues.

The authors have now included two hypotheses. this is commendable>

1. Green innovation positively relates to organizational climate.

2. Green innovation positively relates to organizational synergy

Beyond the relationships no indication of the predictive capacity of the dimensional capacity. Ironically, control variables (region, age, size, and type) are used to predict green innovation whereas no dimensional construct or independent variable was used to predict green innovation. I suggest that the authors go beyond control variables as predictors
